# Cofilactin filaments regulate filopodial structure and dynamics in neuronal growth cones

Ryan K. Hylton [1], Jessica E. Heebner[1], Michael A. Grillo[1] & Matthew T. Swulius [1✉]

Cofilin is best known for its ability to sever actin filaments and facilitate cytoskeletal recycling inside of cells, but at higher concentrations in vitro, cofilin stabilizes a more flexible, hyper-twisted state of actin known as "cofilactin". While this filament state is well studied, a structural role for cofilactin in dynamic cellular processes has not been observed. With a combination of cryo-electron tomography and fluorescence imaging in neuronal growth cones, we observe that filopodial actin filaments switch between a fascin-linked and a cofilin-decorated state, and that cofilactin is associated with a variety of dynamic events within filopodia. The switch to cofilactin filaments occurs in a graded fashion and correlates with a decline in fascin cross-linking within the filopodia, which is associated with curvature in the bundle. Our tomographic data reveal that the hyper-twisting of actin from cofilin binding leads to a rearrangement of filament packing, which largely excludes fascin from the base of filopodia. Our results provide mechanistic insight into the fundamentals of cytoskeletal remodeling inside of confined cellular spaces, and how the interplay between fascin and cofilin regulates the dynamics of searching filopodia.

[1] Department of Biochemistry and Molecular Biology, Penn State College of Medicine, Hershey, PA, USA. ✉email: mts286@psu.edu

In the developing nervous system, functional neural circuits are constructed via a process of chemically and mechanically induced neurite guidance[1–4]. Here, neurites are directed toward their eventual synaptic partners by the coordination of actin polymerization and depolymerization within the "growth cone" found at their distal tips[1–3,5–9]. When grown in culture, the growth cone is typically fan-shaped with filopodial protrusions at the leading edge, which are connected laterally by a lamellipodial veil made of shorter, branched actin networks[2]. Both of these structures constitute the growth cone's peripheral domain, and are highly enriched in filamentous actin (F-actin) (Fig. 1a)[2,3,5–8]. The growth cone advances and turns by integrating both attractive and repulsive cues from the environment and converting them into signaling cascades that drive remodeling of the actin cytoskeleton[2,3,10–15]. Here, filopodia act as antennae, detecting and responding to extracellular cues, while actin network modification in the lamellipodia moves the membrane and drives the growth cone towards its destination[2,5,16–18].

This motion is largely driven by polymerization of actin at the cell's leading edge where the barbed ends of actin are concentrated[5], and it manifests in live-cell movies as a retrograde flow of actin from the leading edge toward the back of the lamellipodium[5]. Retrograde flow is seen both in filopodial and lamellipodial networks, and is regulated through interactions with the myosin family of motor proteins, which increase flow rates[19,20]. It has also been shown that filopodial extension is inversely correlated with retrograde flow[19].

Remodeling of growth cone actin networks is facilitated by an array of actin-binding proteins[6,21]. Cofilin, for example, is a 19 kDa actin-binding protein whose most widely described function is the severing of F-actin[22–25] and the regulation of F-actin polymerization/depolymerization kinetics[22,25,26]. Previous work has shown that cofilin is necessary for normal neurite outgrowth[6,27,28] and that outgrowth can be increased by its overexpression[29].

Multiple groups have demonstrated, in vitro, that purified cofilin binds to F-actin at a 1:1 molar ratio[30,31] in a cooperative manner[32–35], and the consequence of this binding is the disruption of the DNase-I loop of F-actin[36–39], causing a shortening of the filament's helical pitch (~27 nm crossover length compared to ~37 nm for normal F-actin)[31,39,40]. It is thought that this structural change caused by cofilin binding to F-actin facilitates filament severing[36–39]. Further, it has been shown that cofilin severs actin more efficiently at borders between cofilin decorated and undecorated F-actin, displaying a concentration dependence where lower levels of cofilin lead to accelerated severing[22,35,41–46]. Conversely, cofilin-saturated actin filaments (known as cofilactin) can be stabilized at higher concentrations of cofilin[31,40,47], but it is still not known whether cofilactin filaments play a functional role in dynamic actin networks. Light microscopy and fluorescence anisotropy of in vitro filaments has shown that cofilactin is more compliant in both bending[48] and twisting[49] compared to bare actin, suggesting it could alter the mechanical properties of actin networks.

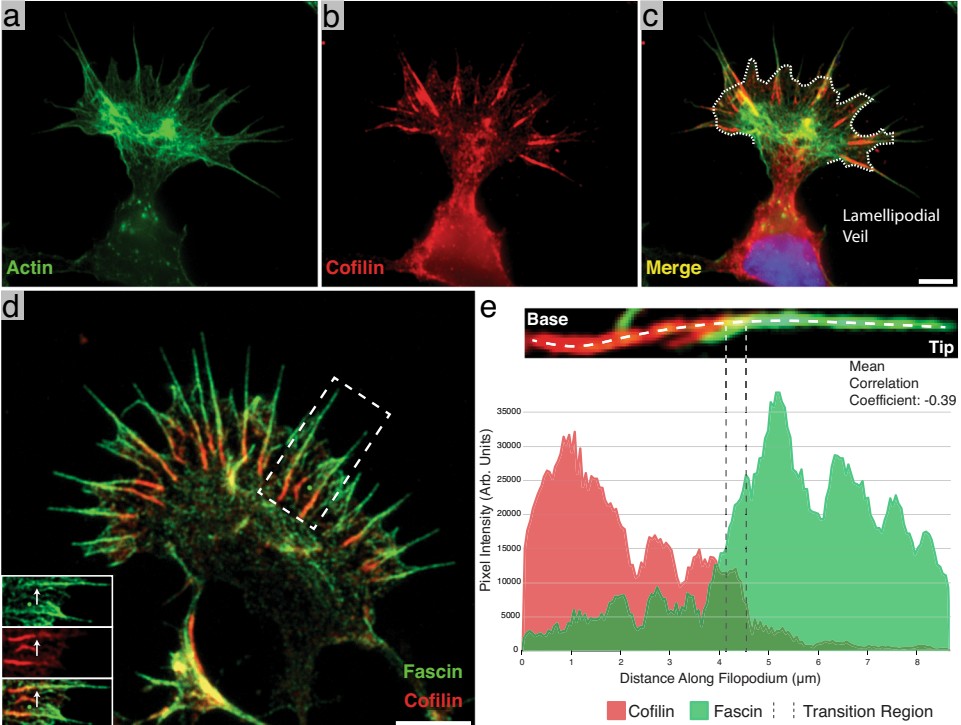

**Fig. 1 Identification of cofilin-rich regions at the base of neuronal growth cone filopodia. a–c** Immunofluorescence images of phalloidin (Alexa Fluor 488) alone (**a**), cofilin (Alexa Fluor 594) alone (**b**), and a merge of the two (**c**). Most of the cofilin signal emanates from linear aggregates near the base of filopodia. The white dashed line in **c** marks the position of the lamellipodial veil. **d** Merged immunofluorescence image of a growth cone with fascin (Alexa Fluor 488-green) and cofilin (Alexa Fluor 594-red) labeled, showing cofilin-rich regions at the base of filopodia as in **c**. Bottom-left inset: Split view of the boxed-out region. The white arrow points to the same location in each image and shows the point at which the fascin signal drops off and the cofilin signal intensifies. **e** Representative line scan intensity profile of a single filopodium showing the distribution of fascin and cofilin. The image above the graph shows a close-up view of the measured filopodium and the location where the line profile was drawn. The transition region is marked by two dashed lines. The images in **a–c** are representative images, but two independent experiments showed similar localization of cofilin and actin. **d** is a representative image from one of three independent experiments. Scale bars: (**c**) 5 µm (this also corresponds to **a** and **b**), **d** 5 µm. Source data are provided as a Source Data file.

Another actin-binding protein, fascin, cross-links actin into filopodia made of hexagonally packed, linear F-actin with regular lateral spacing (~12.3 nm)[50–55]. Previous research in non-neuronal cells suggests that cofilin works synergistically with fascin at the tips of filopodia to sever actin filaments during filopodial retraction[56]. In this model, the twisting cofilactin filaments in the tips of filopodia generate tension against the fascin cross-links, and these localized stress-points increase severing events[56]. However, in neuronal cells, it is unclear how the hyper-twisting of F-actin by cofilin affects fascin-linked bundles of actin, or how those interactions influence growth cone dynamics.

Here we reveal, in neuronal growth cones, that fascin and cofilin inversely localize along filopodial actin bundles, with fascin enriched at the distal tip and cofilin enriched toward the proximal base. The consequence of this gradient is a transition in the structure of filopodial F-actin to cofilactin, with a mixture of the two observed at the regions in between. Additionally, live-cell imaging reveals that filopodial bending occurs primarily at the transition region between the cofilin-rich base and the fascin-rich tip. Our data, combined with findings from the literature, suggest that this structural transition regulates filopodial mobility and, ultimately, neurite outgrowth, by tuning the flexibility of actively searching filopodia as well as regulating interactions with motor proteins, such as myosin II[20,57,58].

## Results

**Cofilin localizes to growth cone filopodia, and forms an inverse gradient with fascin along its length.** In order to determine the distribution of cofilin within the growth cone, we used immunofluorescence (IF) labeling of cofilin and phalloidin staining in rat hippocampal growth cones (DIV 1–3). We observed that, in filopodia, F-actin runs from the tip of the protrusion to within the growth cone body, internal to the lamellipodial veil (Fig. 1a). On approximately half of these filopodial bundles (46%, $n = 1049$ filopodia) cofilin labeling was brightly distributed along the most proximal third of the filopodial bundle length (35.3% ± 1.5 S.E.M., $n = 69$ filopodia) (Fig. 1a–c). We did not observe significant cofilin staining at the tips of growth cone filopodia, but the cofilin-rich region is sometimes seen extending beyond the lamellipodial veil and into the protrusion.

Growth cones double-labeled for fascin and cofilin revealed an inverse gradient with fascin enriched near the distal tip of the filopodia, cofilin enriched near the proximal base, and a transition region (further defined in the Supplementary Notes), where both proteins are similarly expressed and colocalized (Pearson's Correlation Coefficient = 0.39 ± 0.06 S.E.M., $n = 26$ filopodia, Fig. 1d, e). Supplementary Fig. 1 shows representative scattergrams of a whole filopodium and of a transition region. Finally, measuring the length of filopodia both with ($n = 97$) and without ($n = 72$) cofilin at their base revealed that, on average, the fascin-rich regions of filopodia were twice as long when a cofilin-rich base was present (6.1 μm ± 1.6 S.D. vs. 3.3 μm ± 1.6, respectively, Supplementary Fig. 2a), suggesting a role for cofilin in the regulation of filopodial length.

**Filopodial bundles contain cofilactin filaments that rearrange filament packing.** Cryo-electron tomography (Cryo-ET) was used to examine the structure of actin bundles along filopodia in vitrified, cultured wild-type rat hippocampal neurons from their distal tip to their proximal base within the body of the growth cone (Fig. 2a). We found that the distal tips typically contained hexagonally bundled, cross-linked, linear actin filaments, as previously described in other filopodia[50,53] (Fig. 2b, Supplementary Movie 1). Closer to the base of the filopodia we often found hexagonally packed bundles of what appeared to be cofilactin filaments, each with a reduced helical pitch (27.95 nm ± 0.21 S.E.M, $n = 56$ twists,

Fig. 2c, Supplementary Movie 2). The shortened twist of these filaments matched previous reports of cofilactin filament structure determined in vitro[30,31].

Averaging of filament pairs within each region showed that neighboring filaments in the distal tip run parallel to one another, and that their helical pitches are in phase (Fig. 2b, bottom inset). Nearer to the base, however, filaments within cofilactin bundles are out of phase with one another, such that the thick portion of one filament is adjacent to the thin portion of its neighbor (Fig. 2c, bottom inset). To further confirm that these hyper-twisted filaments were cofilactin, subtomogram averages of fascin-linked actin and cofilactin within different bundles were generated and fit with their corresponding atomic models (F-actin, PDB ID: 6T1Y and cofilactin, PDB ID: 3J0S) (Fig. 2d, e). Finally, we measured the interfilament distance within fascin-linked actin bundles (12.3 nm, $n = 9,397$, SD = 2.2 nm) and pure cofilactin bundles (11.5 nm, $n = 7,117$, SD = 2.0), and found that cofilactin filaments are packed 0.8 nm closer to each other (Fig. 2f, g, Supplementary Fig. 3).

From the raw tomograms, we determined how the structure of filopodial bundles is altered by cofilin binding (Fig. 3). It appears that upon cofilin decoration of filopodial actin, the filament pitch is shortened and every other column of filaments rotates around its long axis by 90° with respect to the neighboring column (Fig. 3a). This produces a hexagonally packed bundle of alternating filament orientations, with filaments on the same layer running out of phase with one another, as can be seen directly in the raw data (Fig. 3b). When this configuration of filaments is modeled, it reveals likely steric hindrance between fascin and cofilin binding the same actin monomer (Fig. 4a, b). From the model, one can also measure that, within this cofilactin hexagon, cofilin monomers on neighboring filaments come within 2 nm of each other at their closest atoms, across layers, along the edges that are in phase with one another (Fig. 4c–e). This placement puts them in position to interact through known self-associative properties and potentially cross-link the filament bundle[59,60].

**Filopodia contain segments that are a mixture of F-actin and cofilactin filaments in addition to pure cofilactin bundles.** Filopodial bundles containing both actin and cofilactin (Fig. 5a, b), as well as pure cofilactin bundles (Fig. 5c), were distributed from ~8 microns behind the lamellipodial veil to ~7 microns beyond it, within the membrane-bound protrusion ($n = 14$). From these datasets (Supplementary Movies 3–5), it is clear that both actin and cofilactin coexist within the same filopodial bundle. Figure 5 illustrates this trend by showing three bundles at different locations within different filopodial bundles. In the first tomogram (Fig. 5a), a mixture of actin and cofilactin filaments are seen in a bending (~45°) filopodial protrusion (1.4 microns beyond the veil). In the second tomogram (Fig. 5b), a clear transition from a bundle made of fascin-linked actin to a bundle of cofilactin filaments (1.1 microns behind the veil) can be seen. Fascin-linked actin can be recognized due to its signature horizontal striations (green arrows), while cofilactin can be recognized by the lack of striations and its hyper-twisted appearance (red arrows). In the third tomogram (Fig. 5c), a pure cofilactin bundle can be observed (5.3 microns behind the veil). It is important to note that, both pure fascin-linked bundles (Fig. 2b) and pure cofilactin bundles (as shown in Figs. 2c, 3b, and 5c) tend to be straight, while curvatures in the filopodia are associated with mixtures of both actin and cofilactin (also see below).

**Cofilactin distribution correlates with the dynamics of whole filopodia.** To characterize the dynamics of cofilin-rich bundles in the

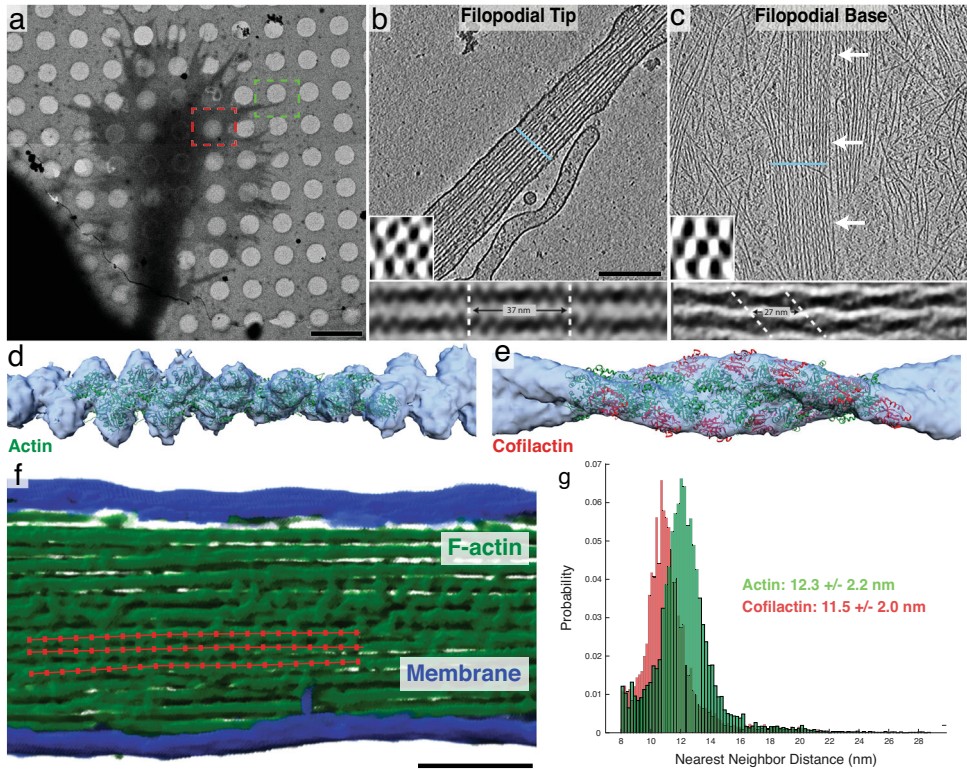

**Fig. 2 Structural features of neuronal growth cone filopodia and their associated cofilactin bundles. a** Overview image of a cryo-preserved growth cone on a Quantifoil EM grid. The green and red boxes represent growth cone regions similar to where the tomograms in **b** and **c** were imaged, respectively. **b**, **c** 5 nm-thick slices of tomograms from the tip (**b**) and base (**c**) of growth cone filopodia. In **b**, a bundle of actin filaments fills the entire cytoplasm. In **c**, branched networks of individual actin filaments can be seen surrounding a central bundle of hyper-twisted cofilactin filaments. White arrows point to the bundle. Lower-left insets: 68 nm-thick transverse cross-sections through each bundle, illustrating the hexagonal packing of filaments. The blue line in the main images show the plane from which the insets were taken. Bottom insets: Subtomogram averages of filament pairs in filopodial tips (below **b**) or in cofilactin bundles at the filopodial bases (below **c**). Cofilactin filaments have a shorter helical twist than F-actin and are out of phase with adjacent filaments. **d** EM map (blue) resulting from the subtomogram averaging of actin filaments in filopodial tips, and rigid body fitting of a previously reported atomic structure for F-actin (PDB ID: 6T1Y; green). **e** EM map (blue) resulting from the subtomogram averaging of cofilactin filaments near the base of filopodia, and rigid body fitting of a previously reported atomic structure of cofilactin (PDB ID: 3J0S; actin is green and cofilin is red). **f** Segmented filopodial protrusion with a schematic of filament centerlines overlaid (red). These lines are comprised of a series of points that were used for nearest neighbor analysis. **g** Nearest neighbor histograms showing the cumulative total of three normal actin filopodial bundles (green) and three cofilactin bundles (red). Scale bars: (**a**) 5 μm, (**b**) 200 nm (this also corresponds to the image in **c**), **f** 100 nm.

growth cone, we collected super-resolution movies (Zeiss Airyscan 2) of rat hippocampal neurons co-expressing fluorescently labeled cofilin and Lifeact (a small peptide that binds F-actin;[61]) (Fig. 6a, Supplementary Movie 6). We observed two basic phenotypes among filopodia in our movies, which we defined as "resting" and "searching" (see Methods for detailed description). In resting filopodia, the cofilin-rich base remained behind the lamellipodial veil, and the bulk of filopodial movement is seen as lateral translations near the proximal base, which creates a smear in the maximum intensity projection (MIP) of our movie frames (Fig. 6b). In the searching phenotype, however, lateral movement of the distal tip, due to filopodial bending, was primarily seen (Fig. 6c).

To classify filopodia into these two groups, we measured the width of the cofilin-rich proximal base, the inflection points at the transition region, and the Lifeact-rich tip within MIPs made from 2-min intervals of our movies. As expected, measurements from resting and searching filopodia revealed an inverse relationship between the maximum widths of cofilin- and Lifeact-rich regions (Supplementary Fig. 2b, c), verifying that the base is more mobile in resting filopodia and the tip is more mobile in searching filopodia. In both cases, however, the inflection points along filopodia were precisely at the transition region, where colocalization of F-actin and cofilin is highest.

Frame-by-frame inspection of our movies showed that movement in resting filopodia was largely produced by lateral shifts and kinks within cofilin-rich filopodial bundles (Fig. 6d, Supplementary Movie 7). These phenomena are known to be driven by Myosin II (MyoII) from experiments in *Aplysia* growth cones[20]. The movement of searching filopodia, on the other hand, produced phenomena such as lateral movement in the rigid filopodial protrusion, which is followed by a wave-like motion that propagates down the cofilin-rich base, or a bending toward the tip as it searches space (Fig. 6e, f, Supplementary Movies 8, 9). Of 79 filopodia examined, 64 (81%) were resting, and 15 (19%) were searching.

Cryotomograms from wild type growth cones reveal the structure of filopodial bundles in each of these dynamic states, and all of them contain cofilactin filaments. For example, Fig. 7a, shows a tomogram from the body of the growth cone, where a ~90° kink can be seen in the actin bundle (Supplementary Movie 10). While fascin-linked actin can be seen before and after the kink, cofilactin filaments are visible in an intermediate region of the bend. In Fig. 7b, a putative filopodial wave was captured just behind the lamellipodial veil (Supplementary Movie 11). Near the top of the image, a fascin-linked actin bundle is seen with a few visible cofilactin filaments embedded within it. Toward

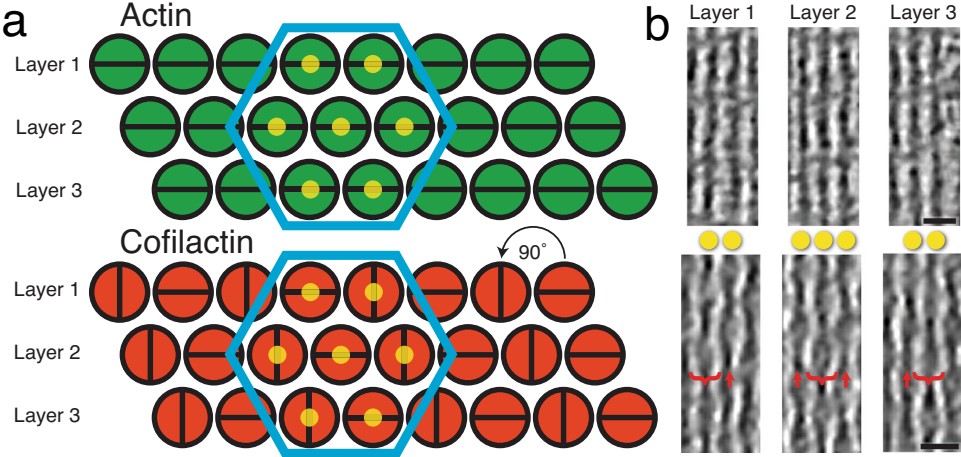

**Fig. 3 Higher-order structure of filopodial bundle types. a** Schematic model of the higher-order structure of actin (fascin-linked) and cofilactin bundles in growth cone filopodia (as they appear looking down the long axis of the bundle). Filaments in both bundle types are organized in layers and hexagonally packed (blue hexagons). Filaments in the actin bundle are all twisting in phase with one another, but in the cofilactin bundle filaments are rotated 90° with respect to their neighbors in the same layer. This creates columns of filaments that are oriented similarly to one another. The yellow dots on filaments in the hexagon correspond to filaments in **b**. **b** 17 nm-thick slices through tomograms of a filopodial tip (top) and a cofilactin bundle (bottom). In the cofilactin bundle, brackets show the wide portion of the helical twist while red arrows show the thin portion. The yellow dots correspond to the dots in **a**. For instance, the bracketed filament in layer two is directly between the two filaments in layers 1 and 3, only on a different *Z*-plane. Scale bars in **b** are 20 nm.

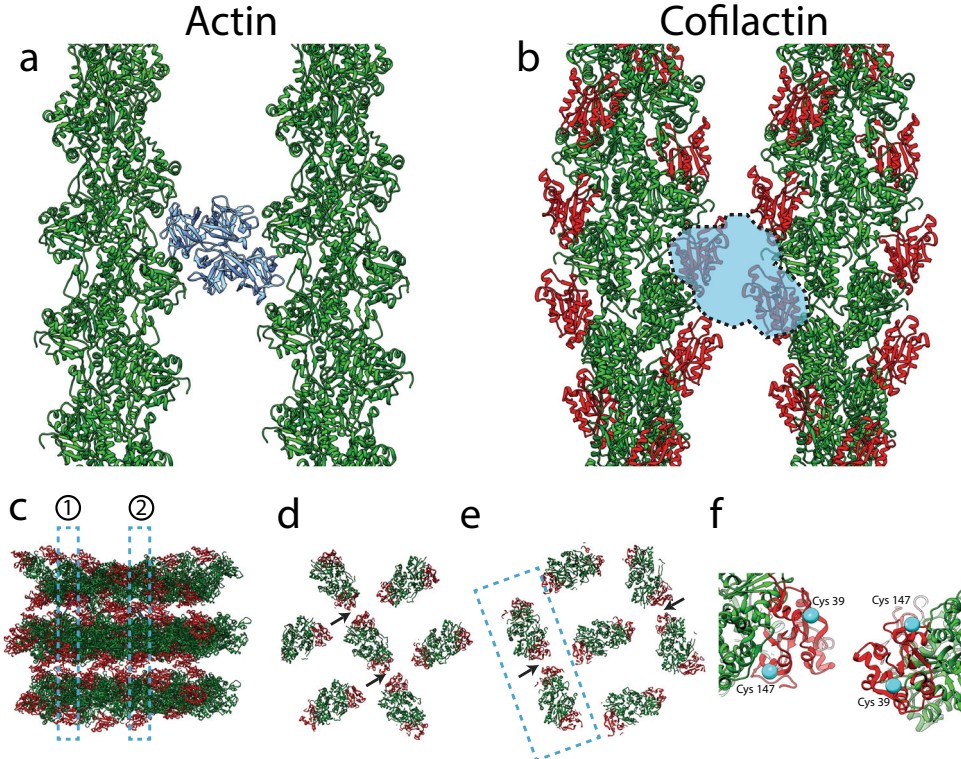

**Fig. 4 Potential conflict between fascin and cofilin, and potential cofilactin interactions. a, b** Top-down views of fascin-linked actin (**a**) and cofilactin (**b**) filaments. Both pairs of filaments are shown at interfaces where they are in phase with one another. In **a**, fascin (blue) is shown in its actin-binding pockets (22), two sites that are sterically blocked by the presence of cofilin in our model of cofilactin bundles (**b**). F-actin is green, and cofilin monomers are red. **c** Sideview of model cofilactin hexagonal unit. Dashed regions labeled 1 and 2 illustrate cross-sections shown in **d** and **e**, respectively. F-actin is green and cofilin is red. **d, e** Cross-sections through different regions of the hexagonal unit shown in **c**. Arrows point to the place where cofilin monomers on neighboring filaments are closest to each other (~2 nm apart at their closest residues). **f** Zoomed-in view of neighboring cofilins like those in the boxed-out region in **e**. Cys39 and Cys147 are displayed on each (cyan).

the bottom of the image there is a curved portion of the bundle where more cofilactin filaments are present. Finally, in Fig. 7c, a filopodial bend is captured, where two bundles (one composed of cofilactin and the other of fascin-linked actin) are presumably splitting in half at the bend point (Supplementary Movie 12).

## Discussion

In this study, we present evidence for the graded structural transition of whole fascin-linked filopodial actin bundles into bundles of cofilactin filaments. This conversion takes place across "transition regions" along the length of filopodia, that can be observed in our live-cell movies and cryo-electron tomograms (Figs. 5–7, Supplementary Movies 3–12). The variable location of this transition region (from 2.6 microns behind the lamellipodial veil to 2 microns beyond it, as measured in our live-cell MIPs, $n = 79$) suggests it could function as a tunable hinge-point that can move from behind the lamellipodial veil into the filopodial protrusion, where it facilitates the searching behavior seen in the tips of a subset of filopodia (Fig. 6c and Supplementary Fig. 2c).

Given that cofilactin is present in these dynamic filopodial structures, especially at regions of curvature, it is logical to think they are involved in regulating the flexibility of filopodial actin bundles. The exact mechanism underlying this flexibility is not obvious from our current data, which revealed a variety of transition region architectures. One possibility is that the increased flexibility of individual cofilactin filaments[48,49] imparts more flexibility to the cytoskeletal network it is bundled within. Another possibility is that the displacement of fascin by cofilin imparts flexibility to the bundle by decreasing lateral cross-links. Most likely, both mechanisms would work synergistically. In our current working model (Fig. 8), highly regular cross-linking of either actin or cofilactin results in a relatively straight and rigid filopodial structure. Mixtures of actin and cofilactin, on the other hand, impart more flexibility through the displacement of fascin cross-linkers as well as a lack of established cofilactin-to-cofilactin interactions. Based on our cryo-ET data, we propose that the displacement of fascin from the transition region is driven through filament twisting by cofilin, which rearranges the alignment between neighboring filaments (Fig. 3). Eventually, as the bundle fully transitions to pure cofilactin filaments, this new registration between neighboring filaments is stabilized by cross-linking between neighboring cofilin molecules or by a currently unknown cross-linker.

Without further experimental evidence, it is difficult to parse whether our observations represent the cause of bending within filopodia, or represent the product of the external forces acting on

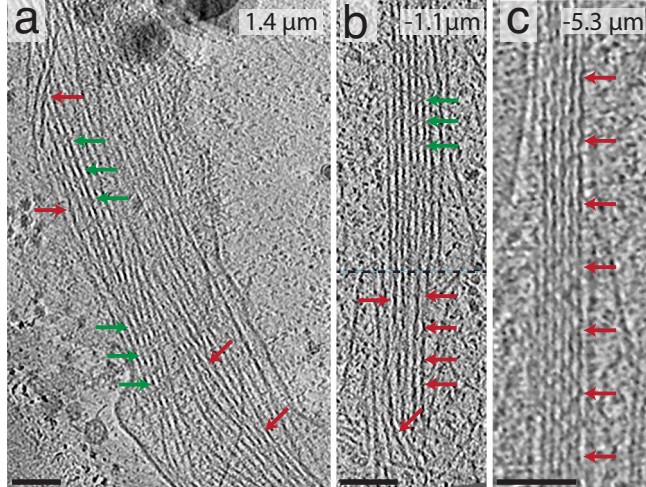

**Fig. 5 Tomography of cofilactin within filopodia.** In all images, red arrows signify cofilactin and green arrows show normal F-actin. The top-right corner of each image shows their distance from the lamellipodial veil (positive values are distal from the veil and negative values are proximal from it). **a** A 13.3 nm-thick tomographic slice where individual cofilactin filaments are scattered throughout a fascin-linked actin bundle. **b** 13.3 nm-thick tomographic slice through a prospective transition region where a clear boundary (represented by the dashed line) exists between the F-actin on top and the cofilactin on bottom. **c** 13.3 nm-thick tomographic slice showing a pure cofilactin bundle). Scale bars: 50 nm.

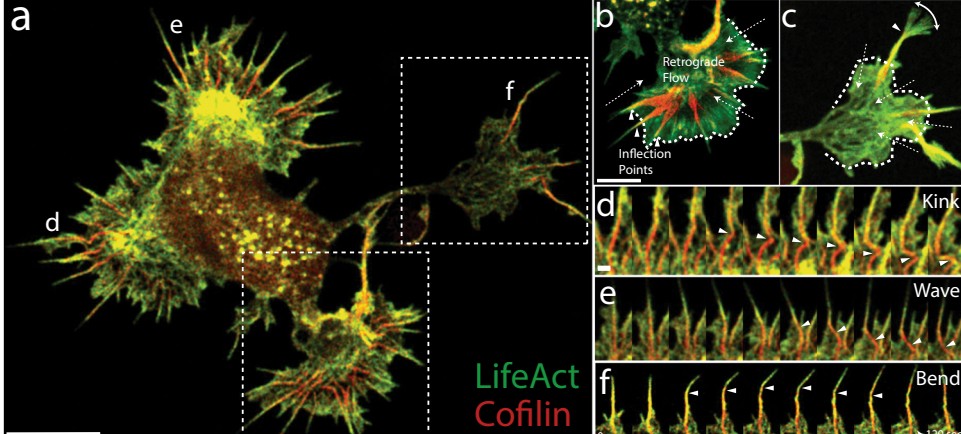

**Fig. 6 Cofilactin bundles facilitate bending and breaking of filopodial protrusions. a** Single image of a whole-cell expressing tdTomato-Lifeact (pseudocolored green) and EGFP-cofilin (pseudocolored red). **b, c** Maximum intensity projections (MIP) of lower and upper boxed regions in **a**, respectively. MIPs include 40 images at 3 s intervals for 2 min total. Filopodia in **b** are examples of "resting" filopodia and (**c**) shows a "searching" filopodium. Dashed arrows indicate the direction of actin retrograde flow, and arrowheads indicate inflection points seen along the flexing filopodial bundles. The white dashed line marks the position of the lamellipodial veil. **d–f** 2-min movie montages showing behaviors exhibited by the cofilin-rich filopodia designated with the corresponding label in **a**. Arrowheads follow either a kink/breaking point (**d**), a cofilactin bundle "wave" (**e**), or an inflection point in a bending filopodium (**f**). In **e**, the wave is caused by the filopodial tip moving from left to right, which drags the attached base behind it in a flexible, wave-like motion. The localization of EGFP-cofilin and tdTomato-Lifeact shown in **a** was replicated in multiple cells from two independent experiments, and similar results were also seen using other fluorescent protein combinations (Supplementary Fig. 5). Scale bars: (**a**, **b**) 5 μm (scale bar in **b** also corresponds to image in **c**), (**d**) 500 nm (also corresponds to (**e**) and (**f**)).

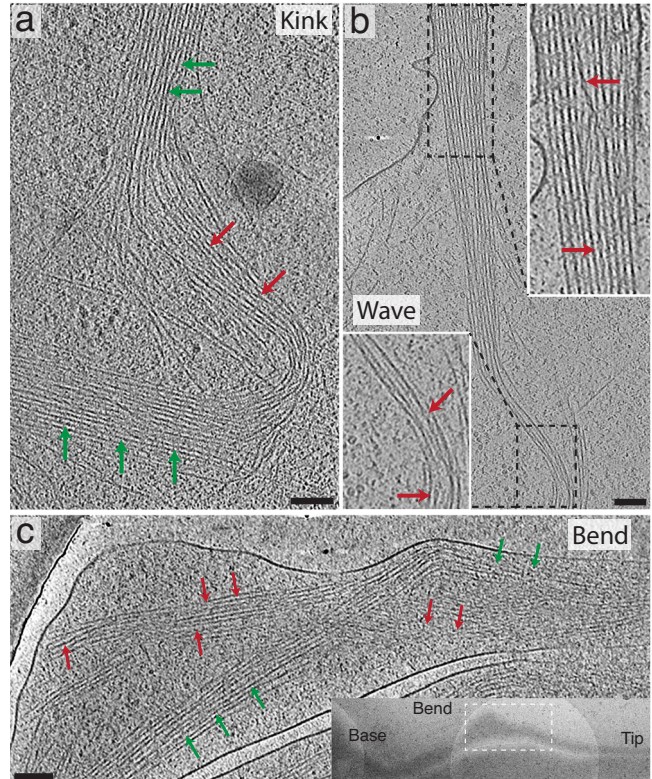

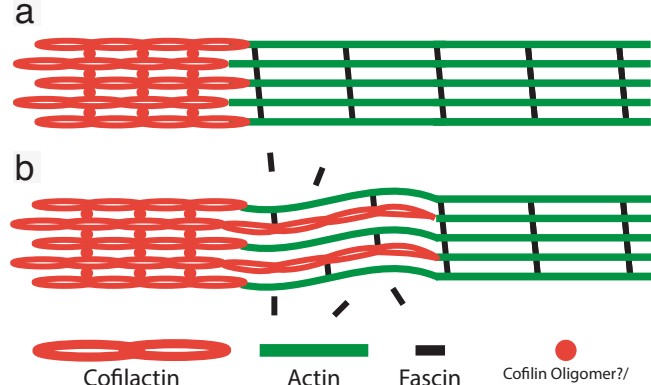

**Fig. 8 Current model.** Schematic of filopodial actin bundle in a rigid (**a**) and flexible (**b**) state. As cofilactin filaments permeate the fascin-linked region of the filament, they competitively dislodge fascin cross-linkers and increase the flexibility of filopodial bundles. In the proximal region of the bundles, cofilactin filaments prevail and are cross-linked through either cofilin oligomers/self-association or by some, as of yet, unknown cross-linker.

**Fig. 7 Tomograms of growth cone filopodia in different dynamic states.** In all images, red arrows signify cofilactin and green arrows show normal F-actin. **a** 13.3 nm-thick tomographic slice showing a ~90° kink in a growth cone filopodium. This filopodium is likely in the process of severing, like in Fig. 6d and Supplementary Movie 7. Cofilactin filaments can be seen near the apex of the kink in the bundle. **b** 13.3 nm-thick tomographic slice where the proximal portion of a filopodium appears to be moving from right to left in a wave-like motion. This is similar to the motion exhibited by the filopodium shown in Fig. 6e and Supplementary Movie 8. The dashed boxes on top and bottom represent the regions shown in the top and bottom zoomed-in insets (white boxes), respectively. The top zoomed-in view shows a cofilactin filament forming an S-curve through the fascin cross-linked bundle. The cofilactin filament is moving through the Z-axis, so the middle of the S-curve is not visible in this image. The bottom zoomed-in view shows, similar to the kink from (**a**), cofilactin filaments near the crest of the wave. **c** 13.3 nm-thick tomographic slice of a bend in a distal filopodial region. Here, cofilactin and actin coexist as separate bundles wrapping around one another. This tomogram resembles that shown in the movie from Fig. 6f and Supplementary Movie 9. Inset: TEM overview image showing the location of the tomogram in the main panel. Scale bars: **a**, **b**, and **c** represent 100 nm.

these cytoskeletal networks. In the case of a filopodial wave, for instance, is it the mixture of actin and cofilactin that allows the wave to prorogate, or is it the prorogation of the wave that generates the mixture? We propose that it's a little of both. It seems equally logical to expect that a network must already be flexible in order for a wave to propagate, yet, it is also easy to imagine that the front of the propagating wave would have an immediate impact on the structure of filopodial networks. There are many open questions still.

Like, what cross-links cofilactin bundles? While our IF data above suggest fascin is minimized within the cofilin-rich base, the fascin signal is still appreciable, (Fig. 1e), and therefore likely plays a role in bundling some filopodial cofilactin filaments, especially in bundles containing a mixture of F-actin and cofilactin. In support of this, a previous study also showed that fascin

fluorescence largely terminates prior to the proximal end of filopodial actin bundles in growth cones[51]. Whatever the major cross-linker of pure cofilactin bundles is, however, it is presumably shorter than fascin, since cofilactin bundles are more tightly packed (Fig. 2g). It's possible that cofilin itself cross-links bundles through self-associative properties[59,60]. Indeed, when cofilactin filaments are arranged according to our model, there is potential for interaction between cofilins on neighboring filaments at the interfaces where their helical twists are in phase with one another. In our model, this in-phase boundary only occurs between filaments on different layers of the bundle, and, at their closest points, neighboring cofilin atoms are ~2 nm apart (Fig. 4c–e). This is too far apart for bonds to form regularly, but based on the range of distances measured by our nearest neighbor analysis (Fig. 2g), a subset of cofilins are close enough to interact within the bundle. Another possibility is that cofilin could oligomerize to span this distance. Indeed, cofilin oligomers have been shown to bundle actin in vitro, while monomers do not[60].

Little is known about the structure of cofilin oligomers, but cysteines 39 and 147 have been implicated in the formation of inter-cofilin disulfide bonds that drive the formation of "cofilin/actin rods", which are rod-like bundles of actin and cofilin that form in neurons under oxidative stress[62–64]. While a subset of these rods persist after the removal of stress, most of them disappear in a reversible process[64]. Based on this, we thought it possible that cofilactin bundles in growth cone filopodia are cross-linked by the same mechanism. If so, it is unlikely that a direct cofilin-to-cofilin bond exists, since the closest pair of cofilins on neighboring filaments puts these residues ~5 nm from each other. Interestingly, however, they are aligned in our model (Fig. 4f), suggesting the possibility of a higher-order oligomeric bridge involving these cysteines and other copies of cofilin. Finally, we think it is a strong possibility that some other cross-linking molecule replaces fascin, but it would need to be able to interact directly with cofilin or with the new regions of actin exposed by cofilin binding.

In addition to oxidation regulating oligomeric structure, it is known that localized oxidative stress impacts cofilin function in growth cones. For instance, MICAL-mediated oxidation of actin has been shown to enhance cofilin binding and severing of actin filaments[65,66]. Moreover, reactive oxygen species (ROS) signaling is necessary for normal growth cone guidance through activation of both MICAL[66] and NADPH oxidase 2 (NOX2)[67]. Given that

transient oxidation of cofilin should enhance its multimerization and its affinity for actin, we propose that it drives an increase its actin-bundling potential. That being said, oxidized actin is more prone to severing by cofilin, but our cryoET data does not reveal small severed fragments of actin in filopodia. Our movies indicate that cofilactin bundles in filopodia are occasionally severed, but typically exist as stable structures for minutes at a time. Future investigation ought to be aimed at elucidating these competing effects of ROS on cofilin/actin dynamics.

Does the cofilactin switch regulate which actin-binding proteins bind to filopodia? Presumably, since this appears to be the case for fascin, but it is also known that cofilin inhibits binding of the Arp2/3 complex to actin[68]. One possibility is that the switch to cofilactin bundles prohibits the base of the filopodia from becoming integrated into the branched lamellipodial actin network, allowing it to move more freely. Indeed, our movies often show the base flexing back and forth quite freely within the growth cone body, or propagating waves along their length (Fig. 6e, Supplementary Movie 8).

Myosin II (MyoII) interactions with filopodia could also be regulated by cofilactin filaments, since cofilin and MyoII are known to compete for binding with actin[57,58], and the MyoII inhibitor blebbistatin has been shown to slow retrograde flow in *Aplysia* growth cones[20]. Given these findings, a switch to cofilactin could reduce retrograde flow by blocking MyoII interactions with filaments within the filopodial base. If so, a reduction in flow rate could account for the fact that filopodia with cofilin-rich regions were twice as long as those without one (Supplementary Fig. 2a), because it has been shown that there is an inverse correlation between retrograde flow and filopodial outgrowth[19]. Also, it is possible that our tomographic example of a splitting bundle within a bending filopodium (Fig. 7c) is the result of differential interactions with myosin motor proteins.

Do cofilactin bundles near the base represent a filopodial breakdown intermediate? They almost certainly do, given cofilin's documented severing function[22–25] and their prevalence near the pointed-end of filopodia, where turnover is known to occur[20]. Indeed, we do observe bundles kinking and collapsing within our movies, near the base of filopodia (Fig. 6d, Supplementary Movie 7), a process known to be energetically driven by non-muscle MyoII[20]. Because of this and other studies showing synergy between cofilin and MyoII[57], we propose cofilactin bundles function to regulate filopodial breakdown in conjunction with the myosin motor proteins. Current evidence suggests that cofilin severs actin filaments by generating intrafilament transitions between cofilactin and bare F-actin. The helical offset between these two portions of the filament is thought to disrupt intermolecular bonds within actin filaments to produce severing[24,41,45,69]. Due to the competitive inhibition exhibited by MyoII and cofilin[57,58], myosin's presence could result in actin filaments that are more sparsely decorated with cofilin, leading to more cofilactin/actin boundaries and more filament severing.

Previous work in non-neuronal cells suggested that cofilin and fascin work synergistically to break down filopodial actin bundles[56]. In their model, the twisting cofilactin filaments at the tip of filopodia compete with the fascin cross-linkers, leading to localized stress and increased severing events. In our model, fascin binding is precluded by the presence of cofilin and the resulting hyper-twisting of actin filaments. Why cofilin and fascin would interact differently in filopodia across cell types is not immediately clear, but given the specialized nature of the growth cone, it could be that this mechanism for cofilactin bundles evolved under the specific pressures of neurite navigation.

It is also possible that methodological differences are to blame. Breitsprecher et al.[56] show cofilin localized to the tip of retracting filopodia of non-neuronal cells by immunofluorescence and using EGFP-tagged cofilin in live-cell movies. From an IF perspective, the fixation and permeabilization protocol used in our studies differs significantly from previous reports. This could, in part, explain the lack of cofilin at filopodial bases in Breitsprecher et al., as we have shown that permeabilization using Triton X-100 eliminates cofilactin bundle staining (Supplementary Fig. 4). Additionally, the dynamics of cofilin, actin, and fascin could be different here than in Breitsprecher et al., where EGFP-Fascin and mCherry cofilin were used[56], two tagged proteins that were not used in this study. That being said, we do not think the distribution of cofilin in growth cone filopodia is simply dictated by the presence of a fluorescent tag (despite it being ~1.4x larger than cofilin). In fact, we show cofilin localized to the proximal base of filopodia in neurons transfected with four combinations of fluorescent tags, including the same EGFP-tag used in Breitsprecher et al. (Supplementary Fig. 5).

Needless to say, there is still work to be done to fully understand the role of actin remodeling in filopodial structure and dynamics, as well as the interplay of cofilin and fascin with other actin-binding proteins. The findings presented here, however, point to a complex role for cofilin in regulating filopodial structure and function that does not only include severing. Finally, given the antennae-like nature of filopodia in axon guidance, we hypothesize that the searching behavior afforded by cofilin-induced bending is essential for optimal space-searching, and, therefore, modulates the efficiency of targeted neurite outgrowth.

## Methods

**Neuronal cell culture**. Coverslips and EM grids were coated with 100 µg/mL poly-D-lysine (PDL) (Millipore) overnight. PDL was subsequently rinsed three times with Milli-Q $H_2O$ prior to cell plating. E18 Sprague Dawley rat hippocampi were acquired from BrainBits LLC (Cat No. KTSDEHP) and cultured according to their protocol. Cells were either cultured in NbActiv4 (BrainBits LLC) or Neurobasal (Gibco) plus 2% (v/v) B-27 Supplement (Gibco), both containing 1% Penicillin–Streptomycin (Gibco). For immunofluorescence experiments, cells were plated at 20,000 cells/cm[2] on 12 mm German glass coverslips (Electron Microscopy Sciences) (either #1 or #1.5 coverslips were used, depending on the microscope). For cryo-ET, cells were plated at 30,000 cells/cm[2] on 200 mesh gold R 2/2 carbon Quantifoil EM grids (Electron Microscopy Sciences).

**Cell vitrification**. On DIV 1, 2, or 3 (~24–72 h after cell plating), cells were vitrified in liquid ethane using a Vitrobot Mark IV (Thermo Fisher Scientific). Prior to vitrification, 10 nm gold fiducial markers (Ted Pella) coated in 1% BSA were diluted 1:4 or 5 in conditioned culture media and 3 µl of this mixture was added on top of each grid. Grids were then blotted by hand from behind for ~2 s with Whatman filter paper and immediately plunge-frozen in liquid ethane.

**Cryo-ET**. EM was performed on a Thermo Fisher Scientific FEI Titan Krios G3i 300 kV FEG TEM equipped with either a 4k × 4k pixel K2 or 6k × 4k pixel K3 direct electron detector (Gatan). A GIF energy filter (Gatan) with a slit width of 20 kV was used during operation and images were collected in electron counting mode. Magnification was typically 26,000x, corresponding to a pixel size of 4.306 Å on the K2 detector and 3.326 Å on the K3. Defocus was −6 to −8 µm. Each tilt series was collected from −60° to +60° with tilt increments of 1° or 2° between images, generally in a bidirectional tilt scheme. A total electron dose of ~150 electrons/Å[2] was used for each tilt series. Data was collected using the software Tomography (Thermo Fisher Scientific).

**3D tomographic reconstruction, tomographic analysis, and subtomogram averaging**. Tomogram alignment and reconstruction were performed in the IMOD software package[70]. Alignment was executed using a fiducial model if possible; otherwise two iterations of patch tracking were used. Reconstruction was generally done by weighted back-projection. All tomograms shown in the figures of this manuscript were filtered for higher contrast either with a median filter (kernel size of 3 pixels) using the clip function in IMOD or a SIRT-like filter, also in IMOD.

Cofilactin twist-lengths were measured in IMOD and calculated using multiple twists (usually 5–10 per filament; 56 measurements total) on a single filament simultaneously. In total, nine filaments from three bundles were measured.

Dynamo was used for subtomogram averaging of individual actin and cofilactin filaments (as in Fig. 2d, e). For cofilactin averages, particles were placed every 27.6 Å and were rotated −162.1° (the approximate axial rise and twist of each actin monomer in cofilactin[31,39,40]). For actin averages, particles were placed every 27.6 Å and were rotated −166.6°[71]. 3,963 particles were used for cofilactin averages

and 1603 particles were used for actin averages. For subtomogram averaging of filament pairs (as in Fig. 2b, c), the software package PEET was used. Rigid body fitting of atomic structures to our EM maps was done in Chimera using the fitmap command.

**Neural network segmentation, filament centerline extraction, and nearest neighbor analysis of tomograms.** Segmentations were generated using Dragonfly software, Version 2021.1 for Windows (Object Research Systems (ORS) Inc, Montreal, Canada, 2020; software available at www.theobjects.com/dragonfly). Full resolution (4k × 4k or 6k × 4k) tomograms were loaded into Dragonfly and filtered with a histogram equalization filter followed by a 3D Gaussian smoothing filter to boost signal. A small rectangular box containing the feature of interest was selected out of the full tomogram. All voxels within this box were hand segmented as either the feature of interest or negative data. The box was used as a training mask for training a neural network. Using the Deep Learning Tool in Dragonfly, a multi-slice (five slices) U-Net was generated for a two-class semantic segmentation. The multiROI generated from hand segmenting the voxels was used as the training output, the filtered tomogram was assigned as the training input and the mask ROI was used to limit the training to the segmented region. All segmentations required some manual clean-up after AI segmentation. The segmented feature of interest was then exported as a binary TIFF.

Binary images were converted from TIFFs to MRC image stacks using the tif2mrc function in IMOD and were imported into Amira where filament centerlines were extracted using its filament tracing tool[72]. A minimum interfilament distance of 8 nm was assigned, as this is the approximate width of an actin filament, therefore, two filaments centerlines could not be any closer than this. The outputted data possessed coordinates of points along filaments from which nearest neighbor distances were calculated in MATLAB (MathWorks) using a custom script. Interfilament distances >3 standard deviations above and below the mean were eliminated as outliers.

**Modeling.** For Fig. 4, UCSF Chimera (www.cgl.ucsf.edu/chimera/) was used to build a seven-filament unit of hexagonally packed filaments, using both F-actin (PDB ID: 6T1Y) and cofilactin (PDB ID: 3J0S) atomic models. Filaments were placed at distances from one another based on our interfilament distance measurements (12.3 for fascin-linked bundles and 11.5 for cofilactin bundles). The atomic model for fascin (PDB ID: 3P53) was fit in by hand, based on modeling done by Aramaki et al.[50]. For cofilactin bundles, filaments were rotated with respect to each other, based on our observations within raw tomograms (Fig. 3).

**Fixation and immunofluorescence.** If they were to be stained with phalloidin for actin staining, neurons were fixed on DIV 1 with a 4% PFA solution (1-part 16% PFA-Electron Microscopy Sciences, 1-part Milli-Q H$_2$O, 2-parts 2x PBS with 8% sucrose) at room temperature for 10 min. This was followed by a 5-min rinse with 50 mM glycine in 1x PBS and three brief rinses in 1x PBS. Cells were then permeabilized by acetone (pre-chilled to −20 °C) for 1 min, followed by a final 1-min rinse in 1x PBS prior to addition of blocking buffer. The cell in Supplementary Fig. 4a was permeabilized with 0.5% Triton-X 100 for 10 min prior to the addition of blocking buffer. If fascin was being labeled instead of actin, cells were simultaneously fixed and permeabilized with methanol (pre-chilled to −20 °C) in the −20 °C freezer for 20 min. This was followed by three brief 1x PBS rinses as above. See the Supplementary Notes and Supplementary Fig. 4 for more details.

Blocking buffer (2% normal goat serum and 1% w/v BSA in 1x PBS) was added for 15 min prior to antibody labeling. Primary antibodies (diluted in blocking buffer from above, 30-min incubation at room temperature): rabbit anti-cofilin at a 1:1000 dilution (Sigma-Aldrich), mouse anti-fascin at a 1:500 dilution (Santa Cruz Biotechnology). Secondary antibodies (also diluted in blocking buffer from above, 30-min incubation at room temperature): goat anti-rabbit conjugated to Alexa Fluor 594 at a 1:500 dilution (Abcam), goat anti-mouse conjugated to Alexa Fluor 488 at a 1:500 dilution (Abcam). Primary and secondary antibodies were rinsed three times for 5 min per rinse with a rinsing buffer (blocking buffer from above diluted 1:10 in 1x PBS). If phalloidin (conjugated to Alexa Fluor 488, Invitrogen) was used, it was diluted 1:66 in the blocking buffer and added after all other antibody labeling and rinsing steps. Afterwards, phalloidin was rinsed three times in the rinsing buffer for only ~30 s per rinse. All coverslips were rinsed once with 1x PBS and once with Milli-Q H$_2$O prior to being mounted on a slide with a hardset mounting medium containing DAPI (Biotium) and allowed to cure overnight in the dark at room temperature.

**Fluorescent imaging.** Images of growth cones where actin and cofilin were labeled were taken on a DeltaVision Elite Deconvolution widefield microscope equipped with a cooled EM-CCD. Images were taken on either a 100x or 60x oil objective with DAPI, FITC (for Alexa Fluor 488 visualization), and Texas Red (for Alexa Fluor 594 visualization) filters. Growth cones where fascin and cofilin were imaged on a Zeiss Axio Observer equipped with a Colibri 7 light source. Here a 100x oil objective was used with a 110 HE LED filter set and EGFP, mRF12, and DAPI filter settings active. Z-stack images were taken with the automated "optimal" Z-section distance used. For some immunofluorescence, the Apotome 2 was used for optical sectioning of the images. In this case, raw Apotome images were deconvolved using Zeiss' Zen Blue software.

**Analysis of fluorescent images.** Brightness and contrast of images were adjusted using FIJI (https://imagej.net/software/fiji/). Additionally, FIJI was used for measurements of bundle and filopodial lengths (Supplementary Fig. 2a) as well as the quantification of the frequency of cofilin-rich bundles in growth cones. All line scan intensity profile measurements were made in Zeiss' Zen Blue software on a central slice from each Z-stack, where the fluorescence intensity was brightest. For the measurements made along the long axis of filopodia (Fig. 1e), one end of the line was placed at the bottom of the cofilactin portion of filopodia and the other end was placed at their most distal tip, as visualized by fascin staining. A correlation coefficient was then calculated from the resulting intensity line graphs of each filopodium measured.

Colocalization analysis was also performed in the Zen Blue software. To determine an optimum threshold for colocalization analysis, three random growth cones were outlined as regions of interest and the "Costes" function within Zen Blue was used before an average threshold pixel value for each channel was calculated. Then, whole filopodia, or just the "transition region" (defined in the Supplementary Notes), were boxed out. The Pearson's Correlation Coefficient was calculated based on the above threshold values.

**Nucleofection.** Primary E18 rat hippocampal neurons (BrainBits LLC) were dissociated from hippocampal tissue slices according to the company's protocol. After dissociation, cells were nucleofected with a Nucleofector 2b (Lonza) using the company's protocol for rat neuronal cells. Briefly, 1–2 million cells were placed in 100 μL of nucleofection solution with a total of 3 μg of plasmid DNA (when two plasmids were used, 1.5 μg of each plasmid was added). Cells were placed in the nucleofector unit and nucleofected using program G-013. After transfection, cells were plated on PDL-coated, 35 mm, glass-bottom dishes (MatTek). Here PDL was the same concentration as above, but plates were rinsed just once with Milli-Q H$_2$O and dried in a cell culture hood for ~1 h prior to use. Cells were incubated overnight in Neurobasal (Gibco) with 10% FBS (R&D Systems) and 1% Penicillin–Streptomycin (Gibco). After overnight incubation, the media was changed to Neurobasal media with 2% (v/v) B-27 Supplement (Gibco) instead of FBS.

**Live cell imaging.** All movies were made on a Zeiss LSM980 using the Airyscan 2 detector and a 40x dipping objective at room temperature. Excitation laser settings were as follows: tdTomato: 561 nm and EGFP: 488 nm. All movies collected were over a 10-min timescale where images were taken every 3 s. Movies were processed using the Airyscan image processing tool in Zen Blue with default settings.

**Quantification of live cell imaging.** Maximum intensity projections (MIPs) were made from images across 40 time points (2 min-worth) using FIJI. The width of filopodia was then measured in three locations: the widest point of the cofilin-labeled portion, the widest point of the actin-labeled portion (more distal than cofilin), and the point at which the two signals met (the inflection point/transition region as described above, in the Main Text, and in the Supplementary Notes). The width in this case represented the amount that each filopodial region moved from side-to-side during the 2-min time block. "Searching" filopodia were defined as those whose actin width was at least 2-fold greater than their cofilin width in the MIPs.

**Plasmids.** All plasmids were acquired from Addgene and verified by sequencing using the associated primers on the Addgene website. The following plasmids were used: LCK-GFP (Addgene 61099), LCK-mScarlet-1 (Addgene 98821), pEGFP-N1 human cofilin WT (Addgene 50859), tdTomato-Lifeact-7 (Addgene 54528), mEGFP-Lifeact-7 (Addgene 54610), and pmRFP-N1 human cofilin WT (Addgene 50856).

**Reporting summary.** Further information on research design is available in the Nature Research Reporting Summary linked to this article.

## Data availability
The subtomogram averaging data that support the findings of this study are available on the Electron Microscopy Data Bank (EMDB) under the accession codes EMD-26214 (actin average) and EMD-26215 (cofilactin average). Source data are provided as a Source Data file.

## Code availability
Custom MATLAB scripts for measuring interfilament distance are available upon request from the authors.

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

## Acknowledgements

We would like to thank Dr. Neal Waxham for insightful discussion and feedback throughout the writing of this manuscript, and Peter Swulius for help with custom Matlab scripts. We would also like to acknowledge support from the cryo-EM core facilities (RRID: SCR_021178) at the Penn State College of Medicine (Hershey, PA) and at the Penn State University Park (State College, PA) campuses. Funding was provided by the Penn State College of Medicine Department of Biochemistry and Molecular Biology.

## Author contributions

The study was conceptualized by R. K. H., M. A. G., and M. T. S.. Formal analysis was done by R. K. H.. Funding was acquired by M. T. S.. Investigation and methodology were carried out by R. K. H., J. H., M. A. G., and M. T. S.. Project administration was carried out by M. T. S.. Resources were acquired R. K. H., J. H., M. A. G., and M. T. S.. Validation was completed by R. K. H., J. H., M. A. G., and M. T. S.. Visualization was carried out by R. K. H., J. H., M. A. G., and M. T. S.. The original draft was written by R. K. H., M. A. G., and M. T. S., and revisions were written by R. K. H. and M. T. S.. M. T. S. supervised this study.

## Competing interests

The authors declare no competing interests.
