## [Peer Review File · Nature Communications]

REVIEWER COMMENTS

Reviewer #1 (Remarks to the Author):

This manuscript addresses the role of cofilin in regulating filopodial structure and flexibility in neuronal growth cones. The imaging presented in the manuscript provides a convincing demonstration that cofilin is enriched at the base of filopodia and engages with actin to form some 'cofilactin' filaments. Furthermore, the actin cross-linker fascin is conversely enriched more toward the tip of filopodia. Based on live imaging of growth cone filopodia the authors propose that actin bundles can switch from fascin cross-linked actin filaments to cofilactin and that the junction between the two is a potential point of breakage of filopodial structure. Whereas the data presented by the authors is consistent with their model, it falls short of establishing causative relationships.

1: There is no experimental manipulation of the growth cones to test cause and effect. All the experimental data is observational and is potentially consistent with multiple interpretations.

2: The authors make the argument that the actin filaments can be severed at the junction of the actin filaments with cofilactin but the electron microscopy falls short of providing convincing evidence of transitions within a single filament from cofilactin to bare actin (as depicted in Fig 4C). It seems equally probable that the cofilactin filaments are separate from the bare actin filaments.

3: The fascin staining at the base of the filopodia, while diminished, remains segregated with actin filaments, and not dispersed as suggested in Fig 4C. This suggests that some actin filaments are cross-linked by fascin along the entire length of filopodia.

Minor point

The sentence on page 5 'It's also possible that cofilin has cell type specific functions, which seems likely given the nature of their evidence' is hard to interpret. What evidence?

Reviewer #2 (Remarks to the Author):

Summary: Hylton et al. present striking cryoEM tomographs of the actin core of growth cone filopodia showing for the first time, a cofilin-saturated actin filament bundle in the proximal region with a transition zone in which a switch is made to fascin-cross linked F-actin bundle in the distal zone. To obtain these images, as well as perfect the method needed to get clear immunostaining of cofilin-actin bundles, they tested different fixation/permeabilization methods and showed that the cofilin-actin bundles were best maintained using methanol as both a fixative and permeabilization treatment. Indeed, the typical formaldehyde fixation and Triton permeabilization method caused the loss of the cofilin immunostaining as others have reported for cofilin-saturated actin bundles found within neurites of stressed neurons. Thus, this is the first report of cofilin-saturated filaments having a direct role in normal cell dynamics. From the cryoEM pictures and modeling of the filament packing, they distinguish changes in transition zone localization and structural flexibility associated with resting, searching and abrupt directional switching of filopodia. Because cofilin saturated filaments are unable to interact with myosin II (myoII), these results also explain some of the previous findings regarding myoII and filopodial dynamics, especially the mechanism of BDNF-induced filopodial elongation reported by Gehler et al. (*J. Neuroscience* 24, 10741-49, 2004). Although Gehler et al showed myoII inhibition led to filopodial elongation, the BDNF signaling pathway through Rho-kinase resulted in the activation of cofilin, and expression of cofilin (S3A) (active form) enhanced elongation independent of myosin II activity. The characterization of the tightly packed cofilin-actin filaments in the cryoEM tomographs shown here also calls into question

other studies on filopodia that have been done using cofilin tagged with fluorescent proteins, since the tagged protein (25 kDa) might disrupt the normal tight packing of the cofilin-actin bundles and result in mislocalization of the tagged cofilin to more distal regions of the filopodium, as has been reported by others. In short, this manuscript helps advance our understanding of neuronal growth cone behavior.

Critique: There are very few criticisms of the work that is presented since the manuscript is well written and the quality of the images is excellent, probably because of the time put into determining the best preservation method for filopodial structures in which both cryoEM structures are best preserved and cofilin remains localized to the proximal filopodium. In addition, the details of the methods are clearly presented.

However, minor corrections, and improvements in some of the figure presentations are suggested below in the order of their appearance in the manuscript.

1. Abstract, line 2-no coma needed before *in vitro*

2. Abstract line 3: cofilactin (cofilin-saturated actin filaments) have been documented in pathological or cell stress conditions such as in nuclear rods or rods within neurites. These are certainly structural roles- so maybe it would be best to state that a functional role in dynamic cell processes has not been observed.

3. Figure 1C. Blue lettering and presumed blue dashed line on the black background is very difficult to find or read. Suggest color change.

4. Top of page 5: ...distributed along the last third..... nearest to the base. Wouldn't it be better to refer to proximal and distal regions of filopodia rather than the "last third" which then needs defining?

5. Page 5, second paragraph. Figure S2 is not in bold text as are all other figures mentioned in text.

6. Page 6: In figure legend it would be useful to point out the arrows in figure 3D and 3E follow the movement of the bend in the filopodium proximal core bundle (wave) leading to a break in Fig 3D. It helps define what is meant by "wave" on Page 7

7. One experiment that would enhance the results of this study is to follow the waves or bends in cells treated with the myoII inhibitor blebbistatin. Is myoII activity necessary in the transition zone for this behavior and does it participate in the transition to fascin cross-linking since myoII can compete with cofilin for filament binding.

8. Page 8, third line: change "believe" to "propose".

9. Page 8, 6th line: delete "our"

10. Page 9, Discussion, second paragraph. The fact that cofilin molecules can be as close as 2 nm apart between filaments in different layers of the bundle should be presented earlier in the text where the interfilament distances are presented. It would be of value to present in the discussion how tagging cofilin with a fluorescent protein, that is 1.4x larger than cofilin itself, might alter the filament packing and might explain the differences in the distribution of cofilin observed in cells expressing EGFP-cofilin compared to what is seen here. This is also referred to in the second to last paragraph on page 10. While there certainly could be cell-type specific differences in filopodia, it is often best to first examine the differences in methodology to see if the reported differences could be artifactual. Based on the core packing of the TEM images from the Breitsprecher et al paper (ref 43) versus what is observed in this study, disruption of the packing by the EGFP-cofilin or fixation/permeabilization methodology seems more likely.

11. It would also be of interest to know if the packing of cofilactin is altered if growth cones are exposed to oxidative stress (e.g. transient peroxide). As mentioned in this paper, persistent rods isolated from stressed neurons have cofilin dimers linked through a disulfide. Actin itself can also be altered in structure through oxidation (cofilin has a greater effect on dynamics of oxidized actin -see Grintsevich EE, et al., 2016 and 2017 papers) and would exposing these filopodial bundles to oxidative conditions lead to cofilin dimerization? There is ample evidence that ROS production is involved in growth cone actin dynamics, both via MICAL (Grintsevich papers) and NADPH oxidase (NOX2) (Munnamalai et al., J

Neurochem, 2014).

12. Page 10 top paragraph: The studies on myoII pulling forces accounting for half of the retrograde flow were performed on Alesia giant growth cones that were essentially immobile. Would one expect motile growth cones to have greater or less effect of myoII on retrograde flow, which should be decreased if forward cell motion is accounted for.

Revisions List:

Below is a point-by-point response to the reviewers' comments/concerns. We would like to thank the reviewers for their time and honesty in reviewing our manuscript. We believe your feedback was critical to the manuscript's evolution, and hope you agree that our new findings better support a particular model. We also hope that our conclusions are more representative of the ambiguities that still exist. We apologize for not tracking the changes in our manuscript. We initially tried, but things eventually became too confusing for even us to read, given that multiple people were heavily editing a single document.

The manuscript has undergone major revision. At the core of the revisions there are two new cryoET figures (Figures 5 and 7), clearly showing filament transitions, as well as mixtures of cofilactin and actin within dynamic filopodial structures. This additional cryoET provides the necessary support for our current working model, which was also adjusted to better represent what the data can support (Figure 8). Multiple of the remaining figures have been moved around for purposes of logical flow, and one figure has been moved from the supplemental data into the main paper (Figure 4). A new supplemental figure (Figure S5) shows the similarity in distribution of different tagged cofilins within living filopodia, and, finally, the bar graphs that were in our original figures have been replotted as box plots and moved to supplemental data (Figure S2).

Reviewer 1:

- 1) "There is no experimental manipulation of the growth cones to test cause and effect. All the experimental data is observational and is potentially consistent with multiple interpretations."

We agree, and hope the paper better represents this fact now. The third paragraph of the discussion addresses this concern specifically. It's our opinion that the discovery of cofilactin within a dynamic cellular context is of great enough importance to the field to warrant publishing observational findings alone.

We have added multiple new tomography figures (Figures 5 and 7) that draw a clearer picture of what is going on within filopodial bundles with respect to cofilactin distribution. We have updated our model figure (now Figure 8) to reflect this, and have added text to emphasize the ambiguities left within our model.

- 2) "The authors make the argument that the actin filaments can be severed at the junction of the actin filaments with cofilactin but the electron microscopy falls short of providing convincing evidence of transitions within a single filament from cofilactin to bare actin (as depicted in Fig 4C). It seems equally probable that the cofilactin filaments are separate from the bare actin filaments."

We have added a figure where clear transitions within individual filaments can be seen (Figure 5B). We do not know, however, that individual filaments are being

severed within bundles, and have altered all of our language to reflect this. We have revised our model (Figure 8) to better reflect what our tomographic data reveals.

- 3) “The fascin staining at the base of the filopodia, while diminished, remains segregated with actin filaments, and not dispersed as suggested in Fig 4C. This suggests that some actin filaments are cross-linked by fascin along the entire length of filopodia.”

We agree with reviewer 1 that fascin is not completely absent within the cofilin-rich segments of the filopodia. It is clearly more concentrated within the filopodium than in the surrounding cytoplasm. We have changed the paper to reflect this, and suggest that fascin is still present, except, perhaps, in pure cofilactin bundles. This is based on the fact that the signature cross-linking densities within fascin-linked bundles are no longer visible within pure cofilactin bundles (Figure 5B), and the spacing between cofilactin filaments within these bundles is significantly smaller (Figure 2G).

Additionally:

- We have also added a reference to the discussion (Cohan et al., 2001), where the authors show that fascin fluorescence signal terminates more distally than does the F-actin signal in growth cone filopodia, which is consistent with our observations shown in Figure 1D & E.

- 4) (Minor Point): “The sentence on page 5 ‘It’s also possible that cofilin has cell type specific functions, which seems likely given the nature of their evidence’ is hard to interpret. What evidence?”

This sentence was meant to indicate that the Breitsprecher et al. paper showed strong evidence that cofilin localized to filopodial tips and that we have no reason to believe our results are correct and there’s are not. This sentence has been removed, and there is now a more thorough description of their work as it compares to ours in the Discussion section.

Reviewer 2:

- 1) “Abstract, line 2-no comma needed before in vitro.”
The comma was deleted.
- 2) “Abstract line 3: cofilactin (cofilin-saturated actin filaments) have been documented in pathological or cell stress conditions such as in nuclear rods or rods within neurites. These are certainly structural roles- so maybe it would be best to state that a functional role in dynamic cell processes has not been observed.”
The wording has been changed to indicate that cofilactin has not been shown to have a structural role in dynamic cellular processes, specifically.
- 3) “Figure 1C. Blue lettering and presumed blue dashed line on the black background is very difficult to find or read. Suggest color change.”
We have changed the blue lettering and dashed lines to white in both Figure 1C, 3B, and 3C. We have also increased the size of the line to make it easier to see.

- 4) “Top of page 5: ...distributed along the last third..... nearest to the base. Wouldn't it be better to refer to proximal and distal regions of filopodia rather than the “last third” which then needs defining?”

We changed “the last third” to “the most proximal third” and deleted the phrase “nearest to the base”.

- 5) “Page 5, second paragraph. Figure S2 is not in bold text as are all other figures mentioned in text.”

“Figure S2” (which is now Figure S1) has been bolded in the text to match all other figure callouts.

- 6) “Page 6: In figure legend it would be useful to point out the arrows in figure 3D and 3E follow the movement of the bend in the filopodium proximal core bundle (wave) leading to a break in Fig 3D. It helps define what is meant by “wave” on Page 7.”

Thanks for the suggestion. We want to be as clear as possible. Because of this we have changed the term break to “kink” since we do not have EM evidence to support breaking of whole filopodial bundles. Additionally, we have added text in paragraph 3 of the section “Cofilactin distribution correlates with the dynamics of whole filopodia” to better differentiate between waves in the base of filopodia, caused by lateral movement of the filopodial tip in searching filopodia, and kinks, which are caused by lateral movement within the base of resting filopodia. We also have added additional description for the arrows in the legend, so that their purpose is clearer.

- 7) “One experiment that would enhance the results of this study is to follow the waves or bends in cells treated with the myoII inhibitor blebbistatin. Is myoII activity necessary in the transition zone for this behavior and does it participate in the transition to fascin cross-linking since myoII can compete with cofilin for filament binding.”

This is a great suggestion, and one that we are working on. So far, in our hands, blebbistatin has given us variable results during live cell imaging, and we are working to resolve this. Additionally, cryoET data of blebbistatin treated neurons (N=1) suggests that there is far more cofilactin present, and while these results are interesting, they are very preliminary and need to be fleshed out much further prior to publication.

- 8) “Page 8, third line: change “believe” to “propose”.”

The word “believe” was used twice in this paper, and in both cases, it has now been replaced with the word “propose”.

- 9) “Page 8, 6th line: delete “our”.”

This word has been deleted.

- 10) “Page 9, Discussion, second paragraph. The fact that cofilin molecules can be as close as 2 nm apart between filaments in different layers of the bundle should be presented

earlier in the text where the interfilament distances are presented. It would be of value to present in the discussion how tagging cofilin with a fluorescent protein, that is 1.4x larger than cofilin itself, might alter the filament packing and might explain the differences in the distribution of cofilin observed in cells expressing EGFP-cofilin compared to what is seen here. This is also referred to in the second to last paragraph on page 10. While there certainly could be cell-type specific differences in filopodia, it is often best to first examine the differences in methodology to see if the reported differences could be artifactual. Based on the core packing of the TEM images from the Breitsprecher et al paper (ref 43) versus what is observed in this study, disruption of the packing by the EGFP-cofilin or fixation/permeabilization methodology seems more likely.”

The ~2 nm distance between neighboring cofilins is now initially mentioned in the Results section when the interfilament distances are shown. We have brought the supplemental figure (previously Figure S5), which supports this distance, into the main text as Figure 4 in order to emphasize the implications of this finding.

The discussion has been updated to reflect potential problems with tagging small proteins. We do not believe, however, that EGFP-tagged cofilin is problematic, in this case, because cofilin localizes at the base of neuronal filopodia with a either EGFP- or RFP-cofilin (we’ve added this as the new Supplemental figure 5). The fact that Breitsprecher et al. also used EGFP-cofilin in their study leads us to think the difference we see is not purely methodological.

We have added a small section to the Discussion where we speculate about the differences in methodology (2nd and 3rd to last paragraph of the discussion) that may have led to differences in results shown in our paper and Breitsprecher et al., especially with regards to our fixation/permeabilization methodology.

- 11) “It would also be of interest to know if the packing of cofilactin is altered if growth cones are exposed to oxidative stress (e.g. transient peroxide). As mentioned in this paper, persistent rods isolated from stressed neurons have cofilin dimers linked through a disulfide. Actin itself can also be altered in structure through oxidation (cofilin has a greater effect on dynamics of oxidized actin -see Grintsevich EE, et al., 2016 and 2017 papers) and would exposing these filopodial bundles to oxidative conditions lead to cofilin dimerization? There is ample evidence that ROS production is involved in growth cone actin dynamics, both via MICAL (Grintsevich papers) and NADPH oxidase (NOX2) (Munnamalai et al., J Neurochem, 2014).”

Thanks for pointing these studies out to us. They are very relevant. We have added discussion to the paper regarding these findings in the first paragraph of page 7.

We agree it will be interesting to study the effect of oxidative stress on filament packing, and to also compare them to cofilin/actin rods. Perhaps the tightly packed bundles of pure cofilactin near the base of filopodia are formed after actin-oxidation by MICAL? We plan to perform such experiments in the future.

12) “Page 10 top paragraph: The studies on myoII pulling forces accounting for half of the retrograde flow were performed on *Aplysia* giant growth cones that were essentially immobile. Would one expect motile growth cones to have greater or less effect of myoII on retrograde flow, which should be decreased if forward cell motion is accounted for.”

Yes, advancing growth cones exhibit reduced retrograde flow, in large part due to the engagement of molecular clutches at focal adhesion contacts, and it may not be appropriate to cite numbers from these immobile *Aplysia* growth cones in this context. We were trying to make a case for myosin’s involvement in retrograde flow, and the specific percentage of effort from myosin is not necessarily critical. We have reworked the discussion around this topic to reflect this. We think the effect of MyoII within motile growth cones would be less pronounced, but would likely still have an effect.

REVIEWERS' COMMENTS

Reviewer #1 (Remarks to the Author):

This paper demonstrates that filopodia emanating from the growth cone contain actin filaments that transition from those cross-linked by fascin to cofilactin filaments with altered pitch that largely eliminate fascin cross-linking. The new figures are excellent and very clearly show this transition. This change in actin correlates very closely with the physical behaviour of the filopodia. While lacking direct experimental manipulation, the results are strongly supportive of the authors' model. This is a significant contribution to the understanding of growth cone and more generally to the structural properties of filopodia.

REVIEWERS' COMMENTS

Reviewer #1 (Remarks to the Author):

This paper demonstrates that filopodia emanating from the growth cone contain actin filaments that transition from those cross-linked by fascin to cofilactin filaments with altered pitch that largely eliminate fascin cross-linking. The new figures are excellent and very clearly show this transition. This change in actin correlates very closely with the physical behaviour of the filopodia. While lacking direct experimental manipulation, the results are strongly supportive of the authors' model. This is a significant contribution to the understanding of growth cone and more generally to the structural properties of filopodia.

Thank you! We appreciate all of your feedback, as it has made the paper much stronger.